# Spike Sorting using the Neural Clustering Process

Yueqi Wang, Ari Pakman, Catalin Mitelut, JinHyung Lee, Liam Paninski

Columbia University

## Abstract

We present a novel approach to spike sorting for high-density multielectrode probes using the Neural Clustering Process (NCP), a recently introduced neural architecture that performs scalable amortized approximate Bayesian inference for efficient probabilistic clustering. To optimally encode spike waveforms for clustering, we extended NCP by adding a convolutional spike encoder, which is learned end-to-end with the NCP network. Trained purely on labeled synthetic spikes from a simple generative model, the NCP spike sorting model shows promising performance for clustering multi-channel spike waveforms. The model provides higher clustering quality than an alternative Bayesian algorithm, finds more spike templates with clear receptive fields on real data, and recovers more ground truth neurons on hybrid test data compared to a recent spike sorting algorithm. Furthermore, NCP is able to handle the clustering uncertainty of ambiguous small spikes by GPU-parallelized posterior sampling. The source code is publicly available [1].

## 1 Introduction

Large-scale neuronal population recordings using high-density multi-electrode arrays (MEA) are at the forefront of current progress in understanding neural circuit dynamics. In MEA recordings, each electrode channel reads extracellular signals from many neurons, and each neuron is recorded by multiple nearby electrodes. A key step in the analysis of MEA data is spike sorting, which converts the raw electrical signal into a set of neural spike trains belonging to individual neurons. As MEAs grow in scale and popularity, there is a new urgency in improving spike sorting performance [2–7].

A typical spike sorting pipeline consists of three steps. The spike detection step extracts putative spike events from noisy recordings. The clustering step groups similar spike waveforms into clusters, each representing a putative neuron. To resolve colliding waveforms, a deconvolution step is often performed. Spike clustering is at the core of the pipeline, as the clustering performance determines both the accuracy of spike assignment and the quality of spike templates used for deconvolution.

Spike clustering, however, poses significant challenges: (1) Spike waveforms form highly non-Gaussian clusters in spatial and temporal dimensions, and it is unclear what are the optimal features for clustering. (2) It is unknown *a priori* how many clusters there are. (3) Although existing methods perform well on spikes with high signal-to-noise ratios (SNR), there remain significant challenges in the lower-SNR regime with increased clustering uncertainty. Fully-Bayesian approaches proposed to handle this uncertainty [8, 9] do not scale to large datasets due to expensive Gibbs sampling.

To address these challenges, we propose a novel approach to spike clustering using the recently introduced Neural Clustering Process (NCP) [10, 11] (Figure 1). NCP is based on a neural architecture that performs scalable amortized approximate Bayesian clustering. (1) Rather than selecting arbitrary features for clustering, the spike waveforms are encoded with a convolutional neural network (ConvNet), which is learned end-to-end jointly with the NCP network to ensure optimal feature encoding. (2) Using a variable-input softmax function, NCP is able to compute full posterior distributions on

Workshop on Real Neurons & Hidden Units: Future directions at the intersection of neuroscience and artificial intelligence, 33rd Conference on Neural Information Processing Systems (NeurIPS 2019), Vancouver, Canada.

cluster labels and the number of clusters, without assuming a fixed or maximum number of clusters. (3) NCP allows for efficient probabilistic clustering by GPU-parallelized posterior sampling, which is particularly useful for handling the clustering uncertainty of ambiguous small spikes. (4) The computational cost of NCP training can be highly amortized, since neuroscientists often sort spikes form many statistically similar datasets.

We trained NCP for spike clustering using synthetic spikes from a simple yet effective generative model that mimics the distribution of real spikes, and evaluated the performance on labeled synthetic data, unlabeled real data, and hybrid test data with partial ground truth. We show that using NCP for spike sorting provides high clustering quality, matches or outperforms a recent spike sorting algorithm [2], and handles clustering uncertainty by efficiently producing multiple plausible clustering configurations. These results show substantial promise for incorporating NCP into a production-scale spike sorting pipeline.

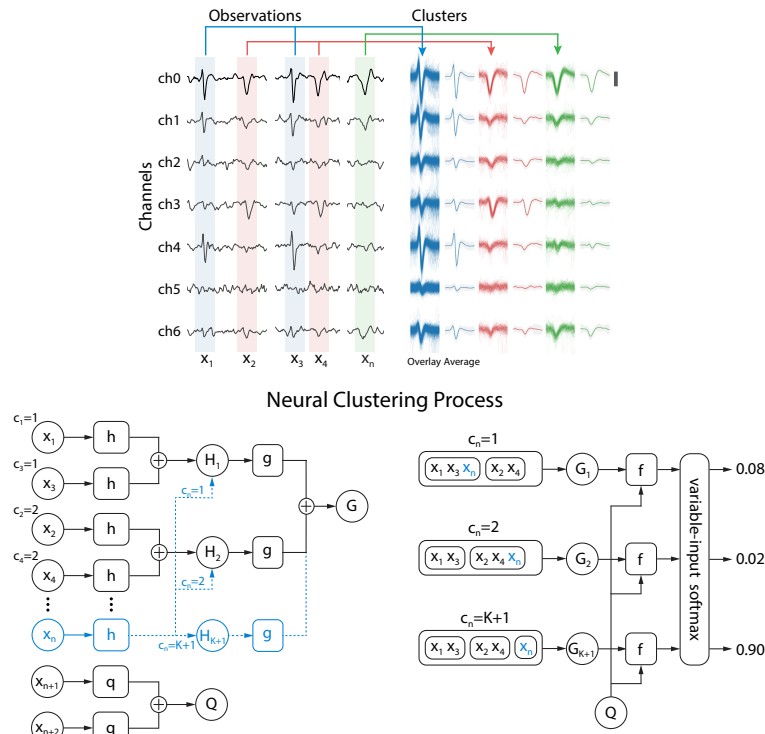

Figure 1: **Spike sorting with NCP.** *Top:* Multi-channel spike waveforms are grouped into clusters by NCP. Each row is an electrode channel. Spikes with the same color belong to the same cluster. (Scale bar: $5\times$ standard deviation (SD)). *Bottom:* Diagram of the NCP architecture as in [11]. The model is composed by the deep networks $h, g, q, f$. *Bottom left:* After assigning the cluster labels $c_{1:n-1}$, each possible discrete value $k$ for $c_n$ gives a different symmetry-invariant encoding of $x_{1:n}$ into the vector $G_k$, using the functions $h$ and $g$. The remaining, yet-unassigned points $x_{n+1:N}$ are encoded by $q$ and summed into the vector $Q$. *Bottom right:* Each pair $G_k, Q$ is mapped by $f$ into a real number (logit), which in turn is mapped into the multinomial distribution $q_\theta(c_n|c_{1:n-1}, \mathbf{x})$ via a variable-input softmax.

## 2 Spike Sorting using the Neural Clustering Process

**Data preprocessing.** Training and test data come from the retinal recordings in [12] using a 512-channel 2D hexagonal MEA with 20 kHz sampling rate. After spike detection [5], each multi-channel spike waveform was assigned to the channel where the waveform has the maximum peak-to-peak (PTP) amplitude (i.e. the center channel, ch0). This partitioned the recording data by channel such that each center-channel-based partition only contains multi-channel spike waveforms centered at that channel. Each spike waveform is represented as a $7 \times 32$ array containing the 32 time steps surrounding the peak from the center channel and the same time window from the 6 immediate

neighbor channels (Figure 1 top). These $7 \times 32$ arrays are the spikes on which clustering was performed.

**Neural architecture for NCP spike sorting.** The NCP architecture contains four neural networks, $h, q, g, f$, as shown in Figure 1 (bottom). We refer to [11] for the detailed formulation and notations of NCP. To extract useful features from the spatial-temporal patterns of spike waveforms, we use a 1D ConvNet as the $h$ and $q$ encoder functions. The convolution is applied along the time axis, with each electrode channel treated as a feature dimension. The ConvNet uses a ResNet architecture with 4 residual blocks, each having 32, 64, 128, 256 feature maps (kernel size = 3, stride = [1, 2, 2, 2]). The last block is followed by an averaged pooling layer and a final linear layer. The outputs of the ResNet encoder are the $h_i$ and $q_i$ vectors of NCP, i.e. $h_i = \text{ResNetEncoder}(x_i)$. We used $d_h = d_q = 256$. The other two functions, $g$ and $f$, are multilayer perceptrons identical to those in the 2D Gaussian example in [11].

**Training NCP using synthetic data.** To train NCP for spike clustering, we created synthetic labeled training data (Figure 2) using a mixture of finite mixtures (MFM) generative model [13] of noisy spike waveforms that mimics the distribution of real spikes:

$$N \sim \text{Uniform}[N_{min}, N_{max}] \quad (1) \qquad c_1 \ldots c_N \sim \text{Cat}(\pi_1, \ldots, \pi_K) \qquad (4)$$
$$K \sim 1 + \text{Poisson}(\lambda) \quad (2) \qquad \mu_k \sim p(\mu) \quad k = 1 \ldots K \qquad (5)$$
$$\pi_1 \ldots \pi_K \sim \text{Dirichlet}(\alpha_1, \ldots, \alpha_K) \quad (3) \qquad x_i \sim p(x_i | \mu_{c_i}, \Sigma_s \otimes \Sigma_t) \quad i = 1 \ldots N \quad (6)$$

Here, $N$ is the number of spikes between $[200, 500]$. The number of clusters $K$ is sampled from a shifted Poisson distribution with $\lambda = 2$ so that each channel has on average 3 clusters. $\pi_{1:K}$ represents the proportion of each cluster and is sampled from a Dirichlet distribution with $\alpha_{1:K} = 1$. The training spike templates $\mu_k \in \mathbb{R}^{7 \times 32}$ are sampled from a reservoir of 957 ground-truth templates not present in any test data, with the temporal axis slightly jittered by random resampling. Finally, each waveform $x_i$ is obtained by adding to $\mu_{c_i}$ Gaussian noise with covariance given by the Kronecker product of spatial and temporal correlation matrices estimated from the training data. This method creates spatially and temporally correlated noise patterns similar to real data (Figure 2). We trained NCP for 20000 iterations on a GPU with a batch size of 32 to optimize the NLL loss by the Adam optimizer [14]. A learning rate of 0.0001 was used (reduced by half at 10k and 17k iterations).

**Probabilistic spike clustering using NCP.** At inference time, we fed the 7 x 32 arrays of spike waveforms to NCP, and performed GPU-parallelized posterior sampling of cluster labels (Figure 1). Using beam search [15, 16] with a beam size of 150, we were able to efficiently sample 150 high-likelihood clustering configurations for 2000 spikes in less than 10 seconds on a single GPU. After clustering, we obtained a spike template for each cluster as the average shape of the spike waveforms. The clustering configuration with the highest probability was used in most experiments.

## 3 Experimental Results

We compared NCP spike sorting against two other methods: (1) Variational inference on a Gaussian Mixture of Finite Mixtures (**vGMFM**) [17], which is an alternative Bayesian clustering algorithm, and (2) **Kilosort**, a state-of-the-art spike sorting pipeline described in [2]. For vGMFM, the first 5 principal components of the spike waveforms from each channel were used as the input features. For Kilosort, we run the entire pipeline using the Kilosort2 package [18].

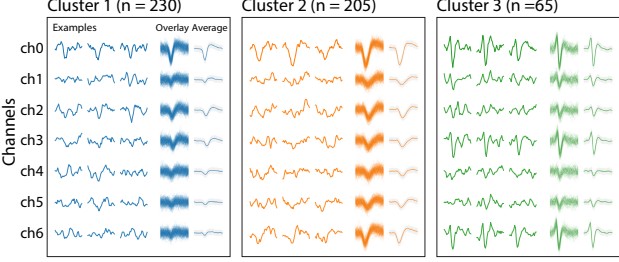

Figure 2: **Synthetic data example.** A synthetic data example containing 3 clusters and 500 spikes.

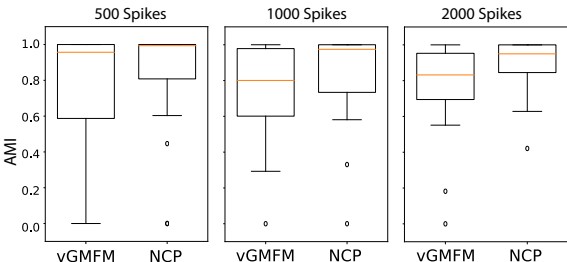

Figure 3: **Clustering on synthetic data.** The AMI scores for 20 sets of 500, 1000, and 2000 unseen synthetic spikes.

**Synthetic Data.** We run NCP and vGMFM on 20 sets of synthetic test data each with 500, 1000, and 2000 spikes. As the ground-truth cluster labels are known, we compared the clustering quality using Adjusted Mutual Information (AMI). As shown in Figure 3, The AMI of NCP is on average 11% higher than vGMFM, showing better performance of NCP on synthetic data.

**Real Data.** We run NCP, vGMFM and Kilosort on a 49-channel, 20-minute retina recording with white noise stimulus, and extracted the averaged spike template of each cluster (i.e. putative neuron). For NCP and vGMFM, we performed clustering on 2000 randomly sampled spikes from each channel (clusters containing less than 20 spikes were discarded), and assigned all remaining spikes to a cluster based on the L2 distance to the cluster centers. Then, a final set of unique spike templates were computed, and each detected spike was assigned to one of the templates. Example clustering results of NCP and vGMFM in Figure 4 (top and bottom-left) show that NCP produces clean clusters with visually more distinct spike waveforms compared to vGMFM.

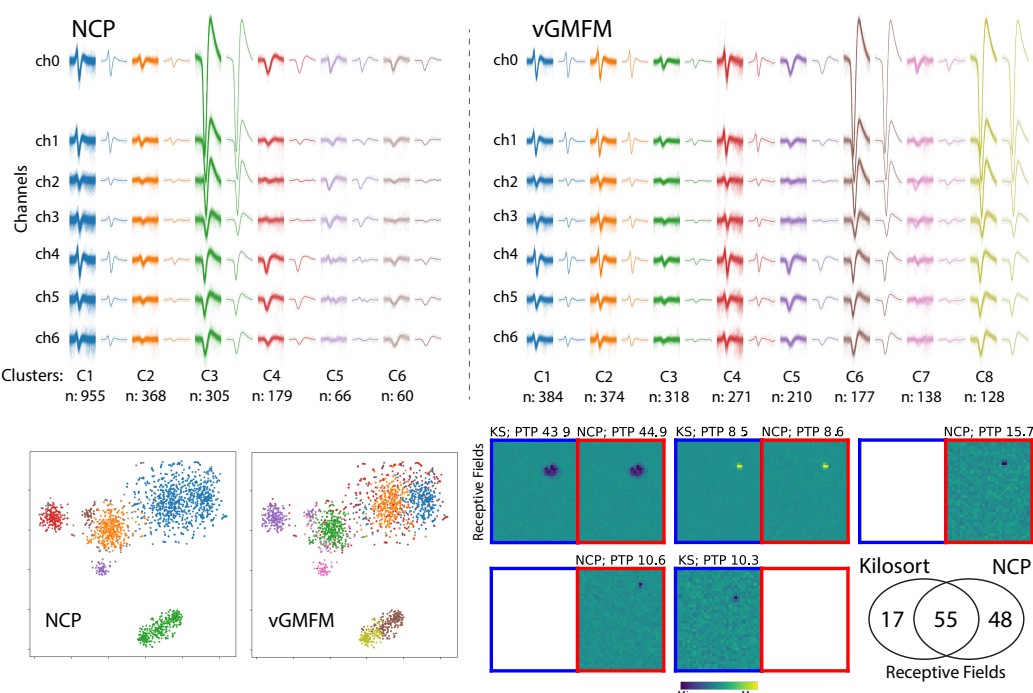

Figure 4: **Spike sorting on real data.** 2000 spikes from real data were clustered by NCP (*top-left*) and vGMFM (*top-right*). Each column shows the spikes assigned to one cluster (overlaying traces and their average). Each row is one electrode channel. *Bottom-left:* t-SNE visualization of the spike clusters. *Bottom-right:* Example pairs of matched retinal RFs recovered by NCP (red boxes) and Kilosort (KS, blue boxes). Blank indicates no matched counterpart. The Venn diagram shows the number of RFs recovered by NCP and Kilosort.

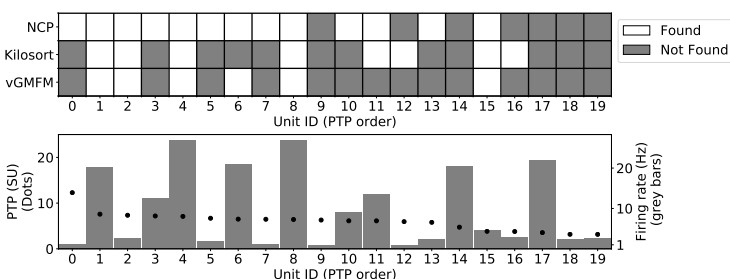

Figure 5: **Spike sorting on hybrid data.** *Top:* NCP, Kilosort, vGMFM recovered 13, 8, and 6 of the 20 injected ground-truth templates. *Bottom:* Peak-to-peak (PTP) size and firing rate of each injected template. (Smaller templates with lower firing rates are more challenging.)

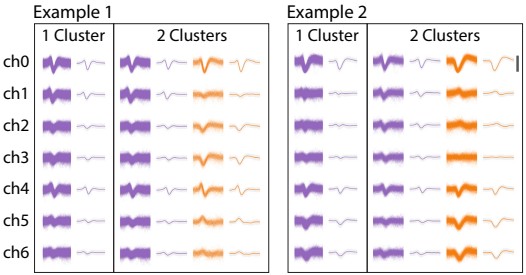

Figure 6: **Clustering ambiguous small spikes.** In both examples, multiple plausible clustering results of small spikes were produced by sampling from the NCP posterior. (scale bar = 5× SD)

As real data do not come with ground-truth cluster labels, we compared the receptive fields (RFs) extracted by NCP and Kilosort; the RF is computed for each cluster as the spike-triggered average of the stimulus (spatiotemporal white noise in this experiment). A clearly demarcated RF provides encouraging evidence that the spike template corresponds to a real neuron. After extracting spike templates and RFs from each pipeline, we matched pairs of templates from different methods by L-infinity distance and pairs of RFs by cosine distance. Side-by-side comparisons of 5 example RF pairs are shown in Figure 4 (bottom-right). See Figure 7 in Appendix for more examples. Overall, the NCP pipeline found 103 templates with clear RFs, among which 48 were not found in Kilosort. Kilosort found 72 and 17 of them were not found by NCP (Figure 4 bottom-right). This shows that NCP performs at least as well as Kilosort, and finds many additional spike templates with clear RFs.

**Hybrid Data.** We compared NCP against vGMFM and Kilosort on a hybrid recording with partial ground truth as in [2]. 20 ground-truth spike templates were manually selected from a 49-channel test recording and injected into another test dataset according to the original spike times. This approach tests the clustering performance on realistic recordings with complex background noise and colliding spikes. As shown in Figure 5, NCP recovered 13 of the 20 injected ground-truth templates, outperforming both Kilosort and vGMFM, which recovered 8 and 6, respectively.

**Probabilistic clustering of ambiguous small spikes.** Spike sorting of small-amplitude waveforms has been challenging due to the low SNR and increased uncertainty of cluster assignment. Traditional methods could not handle the uncertainty and previous fully-Bayesian approaches do not scale. By efficient GPU-parallelized sampling of cluster labels from the posterior, NCP is able to handle the clustering uncertainty by producing multiple plausible clustering results. Figure 6 shows examples where NCP separates spike clusters with amplitude as low as 3-4× the standard deviation of the noise into plausible units that are not mere scaled version of each other but have distinct shapes on different channels.

**Conclusions.** Our results show that NCP spike sorting achieves high clustering quality, matches or outperforms a state-of-the-art method, and is able to handle clustering uncertainty by efficient posterior sampling. Future directions include more realistic generative models, better spike encoders that utilize information from distant channels, and integrating NCP into a standard spike sorting pipeline.

## Acknowledgements

We thank E.J. Chichilnisky, Nora Brackbill, Georges Goetz, Nishal Shah, and Alexandra Tikidji-Hamburyan for many helpful conversations. We also thank the developers of Kilosort/Kilosort2 for generously sharing their code. This work was partially supported by the Simons Foundation, the ONR, and NSF grants IIS-1546296 and NSF IIS-1430239. This research was also developed with funding from the Defense Advanced Research Project Agency (DARPA), Contract No. N66001-17-C-4002. The views, opinions and/or findings expressed are those of the author and should not be interpreted as representing the official views or policy of the Department of Defense of the U. S. Government.

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

# A   Receptive Fields

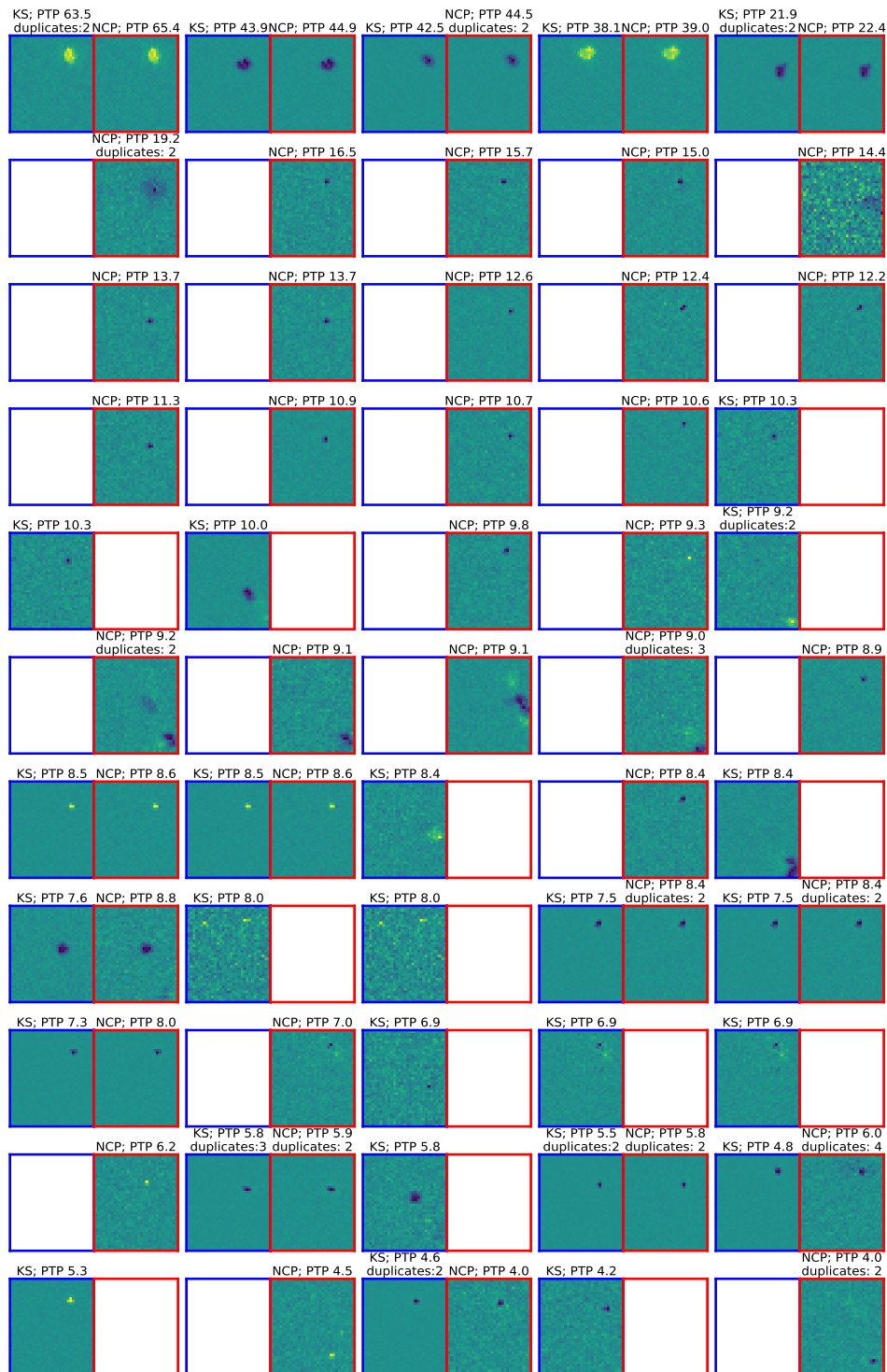

Figure 7: **Comparison of receptive fields extracted by NCP and Kilosort.** Receptive fields of 55 randomly selected pairs of matched units recovered from NCP spike sorting and Kilosort (KS). Red boxes indicate units found by NCP; blue boxes by Kilosort. Blank indicates no matched counterpart. The units are ordered by the PTP size of the spike templates. Both approaches found the spikes with the largest PTP size (first row). For units with smaller PTP, often one sorting method found a cell that the other missed. Overall, NCP found more units with clear receptive fields than Kilosort.

