# OpenReview forum: "Spike Sorting using the Neural Clustering Process"
_NeurIPS.cc/2019/Workshop/Neuro_AI — Real Neurons & Hidden Units @ NeurIPS 2019 Poster_

### Official Review · AnonReviewer1 · 2019-09-24
**Clear & well written, if preliminary :-)**

**Clarity:** 4

**Comment:**

Paper is well written and fairly easy to read. The efficiency of amortised methods seems clear for increasingly large data sets, but perhaps a point of concern is how confident we can be about doing supervised learning for an inherently 'unsupervised' problem?! Training on synthetically labelled data to do inference on true data?

**Category:**

AI->Neuro

**Clarity Comment:**

The text is well written & clear, making the shortfalls of previous methods explicit & being clear about which of these the proposed method aims to address.
Perhaps the data pre-processing section could be a little clearer. A little ambiguous is "We partitioned the recording data such that the data for each channel only contains spikes centered at that channel" but then to also say "For each spike, the extracted waveform is... from the center channel and its 6 immediate neighbor channels." This could perhaps be better phrased?



**Evaluation:**

3: Good

**Importance:**

3: Important

**Importance Comment:**

While spike sorting is probably not the major bottleneck to our effective use & interpretation of large multi-electrode array neural recordings, there is definitely room for improvement on current methods - in performance, versatility & comp. efficiency. Hence the work can be considered marginally important.


**Intersection:**

3: Medium

**Intersection Comment:**

This is clearly a description of trying to optimise machine learning approaches for a well known, 'hard' problem in neuroscience. Hardly see any 'AI' relation.

**Rigor Comment:**

Description of the algorithm is rigorous enough for this setting, as is comparison to other models. Perhaps would be nice to see comparisons to YASS (if possible) since this seems to outperform Kilosort? Similarly since the model is by definition trained on synthetic data, the richness & 'accuracy' of this generative model is key. Hence my label 'preliminary'.

**Technical Rigor:**

3: Convincing

---

### Official Review · AnonReviewer2 · 2019-09-26
**Another spike sorting algorithm based on neural network**

**Clarity:** 4

**Category:**

AI->Neuro

**Clarity Comment:**

The text is well written.

**Evaluation:**

4: Very good

**Importance:**

4: Very important

**Importance Comment:**

Efficient and accurate spike sorting algorithm for multi-channel extracellular recordings is necessary, especially in the low SNR (small spikes) domain.

**Intersection:**

4: High

**Intersection Comment:**

Used AI technique to solve spike sorting problem.

**Rigor Comment:**

Used a previously developed neural network structure to handle neural data. The network structure and logic is very clear. However, it is a bit unclear how well this algorithm perform to reduce the uncertainty of small amplitude spikes compared to other algorithms. It is also a bit unclear whether there is over-splitting of clusters in this algorithm.

**Technical Rigor:**

3: Convincing

---

### Decision · Program_Chairs · 2019-10-02

Accept (Poster)